# Melatonin Rescues Dimethoate Exposure-Induced Meiotic and Developmental Defects of Porcine Oocytes

**DOI:** 10.3390/ani12070832

**Published:** 2022-03-25

**Authors:** Qi Jiang, Xin Qi, Chi Ding, Xingyu Liu, Yuanyuan Lei, Siying Li, Zubing Cao

**Affiliations:** Anhui Province Key Laboratory of Local Livestock and Poultry, Genetical Resource Conservation and Breeding, College of Animal Science and Technology, Anhui Agricultural University, Hefei 230036, China; jq19909845880@163.com (Q.J.); qx13655595865@163.com (X.Q.); dc2071551890@163.com (C.D.); jiuyuandxingyuang@163.com (X.L.); lyy2916684423@163.com (Y.L.); ai20020104@163.com (S.L.)

**Keywords:** dimethoate, melatonin, oocyte maturation, ROS, autophagy

## Abstract

**Simple Summary:**

Environmental pollution poses concerns for public health. Dimethoate is a pesticide widely used in agricultural fields and home gardens. Recent studies have shown that dimethoate exposure impaired reproductive functions in male and female animals. However, whether dimethoate exposure affects oocyte maturation and how to reduce the toxicity of dimethoate remain unclear. Here, we showed that dimethoate exposure impaired nuclear and cytoplasmic maturation of porcine oocytes. Melatonin supplementation restored the meiotic maturation of dimethoate-exposed oocytes by suppressing the generation of excessive reactive oxygen species and autophagy and DNA damage accumulation. Therefore, melatonin counteracts the toxic effects of dimethoate exposure on porcine oocyte maturation. These findings imply that melatonin could be a promising agent in improving the quality of dimethoate-exposed oocytes from humans and animals.

**Abstract:**

Dimethoate (DT) is an environmental pollutant widely used in agricultural fields and home gardens. Studies have shown that exposure to DT causes reproductive defects in both male and female animals. However, the effects of DT exposure on oocyte maturation and the approach to counteract it are not yet known. Here, we investigated the toxicity of DT on porcine oocyte maturation and the protective effects of melatonin (MT) on DT-exposed oocytes. DT exposure with 1.5 mM partially inhibited cumulus cell expansion and significantly reduced the rate of first polar body extrusion (pb1) during oocyte maturation. Parthenogenetically activated embryos derived from DT-exposed oocytes could not develop to the 2-cell and blastocyst stage. Furthermore, DT exposure led to a significant increase in the rates of misaligned chromosomes, disorganized spindles, and abnormal actin assembly. DT exposure severely disrupted the distribution patterns of mitochondria in oocytes but did not change the subcellular localizations of cortical granules. Importantly, MT supplementation rescued the meiotic and developmental defects of DT-exposed oocytes through repressing the generation of excessive reactive oxygen species (ROS) and autophagy, and DNA damage accumulation. These results demonstrate that melatonin protects against meiotic defects induced by DT during porcine oocyte maturation.

## 1. Introduction

The widespread use of pesticides benefits agricultural crops to eradicate various diseases, pests, and weeds. However, pesticide residues in foods, drinking water, and animal feeds pose serious impairments on human and livestock health [1,2,3]. A growing list of evidence suggests that environmental exposure to pesticides frequently causes numerous diseases, such as diabetes, cancer, respiratory, neurodegenerative, and reproductive disorders [4,5,6].

Dimethoate (DT; O,O-dimethyl-S-(*N*-methylcarbamoylmethyl phosphorodithioate) is an organophosphate insecticide extensively used in agricultural fields and home gardens [7]. Humans and animals are chronically exposed to low doses of DT residues that exist in foods, drinking water, and feeds [8,9,10]. It has been reported that DT exposure exerts toxic effects on a variety of organs/cells, in turn causing their dysfunctions. For instance, DT exposure led to injuries of liver [11], lung [12], kidney [13], skeletal muscle cells [14], erythrocytes [15], and pancreatic stellate cells [16] in rats. Exposure to DT impaired the immune system in mice [17]. In addition, DT exposure has adverse effects on reproductive systems. In males, maternal exposure to DT disrupted the pituitary-testicular axis of mice neonates and inhibited steroidogenesis of adult mice [18]. Indeed, direct exposure to DT by oral administration reduced testicular weight and severely impaired sperm quality in mice [19,20] and rats [21]. At the cellular level, DT exposure also inhibited steroidogenesis [22] and differentiation of leydig cells [23]. In females, reductions in both ovarian weight and the number of healthy follicles, irregular estrous cycles, and altered levels of serum hormones have been observed in mice maternally exposed to DT [24]. Moreover, maternal exposure to DT not only reduced the number of both implantations and live fetuses, but also increased the incidences of early embryo resorption [25]. Accumulating data indicated that DT exposure in different cell types and animal models mainly triggered oxidative stress to alter chromatin structure, organelle distribution, and cellular functions [26]. However, the toxic effects and mechanism of DT on oocyte maturation remain unknown.

It is inferred that inhibition of oxidative stress would be an effective approach to reduce the toxicity of environmental pollutants. Studies have indicated that antioxidants can overcome environmental insults-induced oxidative stress in cells [27,28]. Melatonin (MT; *N*-acetyl-5-methoxytryptamine) is a pineal gland-secreted neuro-hormone that possesses an antioxidative ability to scavenge free radical [29]. It was reported that MT ameliorates oxidative stress caused by reactive oxygen species (ROS) to exert protective effects on follicle development, ovulation, oocyte maturation, and embryo development [30,31]. Additionally, emerging evidence has shown that MT can reduce the toxicity of some environmental contaminants, including mycotoxins [32], pesticides [33], plasticizers [34], and heavy metals [35]. Environmental pollutant-induced damages of oocyte quality in mice [36], cattle [37], and pigs [38] could be rescued by MT supplementation. Thus, we speculated that MT could rescue DT-exposure-induced defects during porcine oocyte maturation.

Physiological and developmental characteristics of porcine oocytes are more similar to humans relative to mice [39]. Thus, porcine oocytes in this study were used as a model to assess the toxic and protective effects of DT and MT on meiotic maturation, respectively. We find that DT exposure causes meiotic and developmental defects of porcine oocytes through disrupting the chromosome and spindle organization, actin filament assembly, mitochondria, and cortical granule distribution. In contrast, MT effectively ameliorates the increased oxidative stress, DNA damage, and autophagy, thereby repairing the meiotic defects of DT-exposed oocytes.

## 2. Materials and Methods

All reagents used in this study were purchased from Sigma (Sigma-Aldrich, St. Louis, MO, USA), unless otherwise stated.

### 2.1. Preparation of Dimethoate and Melatonin

DT and MT were dissolved in embryo culture water and DMSO, respectively. Oocyte maturation medium was used to dilute the DT and MT stock solution to obtain the desired working solution. The final concentrations of DMSO in the culture medium were not more than 0.1%.

### 2.2. Oocyte Maturation In Vitro

Ovaries were collected from a local slaughterhouse and transported to the laboratory at 28–35 °C in physiological saline solution. Antral follicles at 3–6 mm in diameter were punctured using a sterile 10 mL syringe. Cumulus-oocyte complexes (COCs) with more than two-layer intact cumulus cells were selected under a stereomicroscope. COCs were subsequently cultured in one well of a 4-well plate containing 400 μL maturation medium (TCM-199 supplemented with 5% FBS, 10% porcine follicular fluid, 10 IU/mL eCG, 5 IU/mL hCG, 100 ng/mL L-Cysteine, 10 ng/mL EGF, 2.03 × 10^−5^ ng/mL LIF, 2 × 10^−5^ ng/mL IGF-1, 4 × 10^−5^ ng/mL FGF2, 100 U/mL penicillin and 100 mg/mL streptomycin) for 42 h at 38.5 °C, 5% CO_2_ and saturated humidity. Hyaluronidase in DPBS without Ca^2+^ and Mg^2+^ (Gibco, Grand Isle, NY, USA) was used to remove the cumulus cells surrounding oocytes.

### 2.3. Oocyte Activation and In Vitro Culture

Oocytes containing pb1 were electronically stimulated with two pulses of direct current (1.56 kV/cm for 80 ms) in activation medium (0.3 M mannitol supplemented with 0.1 mM CaCl_2_, 0.1 mM MgCl_2_ and 0.01% polyvinyl alcohol). Subsequently, embryos were incubated for 4 h in the chemically assisted activation medium (PZM-3 supplemented with 10 μg/mL cycloheximide and 10 μg/mL cytochalasin B). Embryos were then washed three times with PZM-3 medium and cultured in fresh PZM-3 medium for 7 days at 38.5 °C, 5% CO_2_ and 95% air with saturated humidity. The number of cleaved embryos and blastocysts were recorded at day 2 and day 7, respectively.

### 2.4. Immunofluorescence Staining

Oocytes were fixed in 4% paraformaldehyde (PFA) solution for 15 min, permeabilized with 1% Triton X-100 in DPBS for 30 min at room temperature (RT) and then blocked with 2% BSA in DPBS at RT for 1h. The oocytes were co-incubated with the primary antibodies overnight at 4 °C. Following three washes with DPBS containing 0.3% PVP, the oocytes were co-incubated with the secondary antibodies for 1 h in the dark at RT. After three washes, the oocytes were counterstained for 10 min in 4, 6-diamidino-2-phenylindole dihydrochloride (DAPI) solution. Finally, the fluorescence images of oocytes were captured using inverted fluorescence microscope (Olympus, Tokyo, Japan). NIH Image J software was used to calculate the fluorescence intensity of the target proteins in oocytes. Detailed information regarding primary and secondary antibodies used is listed in Appendix A.

### 2.5. Determination of Mitochondrial Distribution

Oocytes containing pb1 were cultured in a maturation medium containing 0.5 μM Mito Tracker TM Red CMX Ros in the dark for 30 min at 38.5 °C, 5% CO_2_ and saturated humidity. The oocytes were washed three times, then fixed on glass slides with coverslips, and imaged using an inverted fluorescence microscope (Olympus, Tokyo, Japan). Fluorescence intensity was calculated with Image J.

### 2.6. Evaluation of Cortical Granules Distribution

Zona-free matured oocytes were fixed in 4% PFA solution for 15 min and then permeabilized with 0.5% Triton X-100 in DPBS for 30 min at RT. Following three washes, the oocytes were incubated with 100 μg/mL FITC-PNA for 30 min at RT. The oocytes were washed three times and counterstained with DAPI for 10 min. Subsequently, fluorescent images were captured with an inverted fluorescence microscope (Olympus, Tokyo, Japan). Fluorescence intensity was calculated with Image J.

### 2.7. Determination of ROS Levels

Matured oocytes were co-incubated with 10 μM DCFH-DA fluorescent probe for 20 min in the dark at 38.5 °C. The oocytes were washed using DPBS containing 0.3% PVP. The fluorescent images were captured with an inverted fluorescence microscope (Olympus, Tokyo, Japan). Fluorescence intensity was calculated with Image J.

### 2.8. Statistical Analysis

At least three replicates were performed for each experiment. Data were statistically analyzed using one-way ANOVA or Student’s t test (SPSS 17.0) and presented as mean ± standard error of mean (mean ± S.E.M). *p* < 0.05 was considered to be statistically significant.

## 3. Results

### 3.1. Dimethoate Exposure Impaired Oocyte Maturation and Early Embryonic Development

To examine the effects of DT exposure on porcine oocyte maturation, GV oocytes were matured in medium supplemented with different concentrations of DT (1 mM, 1.5 mM and 2 mM). As shown in Figure 1A, the expansion status of cumulus cells surrounding oocytes exposed to 1 mM DT was similar to that in the control group. DT exposure with 1.5 mM partially inhibited the cumulus expansion, but DT exposure with 2 mM completely inhibited the cumulus expansion compared to control group. Correspondingly, 1 mM DT exposure did not affect the rate of pb1 extrusion, but DT exposure with 1.5 mM and 2 mM significantly reduced the rate of pb1 extrusion compared to the control group (Figure 1B) (*p* < 0.05). In addition, to examine the developmental competence of DT-exposed oocytes, oocytes with pb1 were parthenogenetically activated and developed to the blastocyst stage. We observed that DT exposure with different concentrations apparently reduced the cleavage rate compared to control group (Figure 1C,D) (*p* < 0.05). The blastocyst rate of oocytes exposed to 1 mM DT significantly decreased compared to control group, and oocytes exposed to 1.5 mM and 2 mM DT could not develop to blastocyst stage (Figure 1C,D) (*p* < 0.05). The concentration of 1.5 mM DT was selected for the subsequent studies because it allows a part of oocytes to be matured for other studies. Therefore, these results indicate that DT exposure inhibited oocyte maturation and early embryonic development.

### 3.2. Melatonin Rescues Meiotic and Developmental Failures of DT-Exposed Oocytes

Given that melatonin (MT) has diverse beneficial effects on oocyte maturation in several species [31], we thus attempted to explore whether melatonin has a protective role against meiotic and developmental defects caused by DT exposure. To test this possibility, different concentrations (0.1 μM and 1.0 μM) of melatonin were supplemented to the maturation medium containing 1.5 mM DT. The results revealed that supplementation of 1 μM MT improved the cumulus cell expansion and restored the rate of pb1 extrusion compared to the DT-exposed group (Figure 2A,B) (*p* < 0.05), but supplementation of 0.1 μM MT did not exhibit the beneficial effects on cumulus expansion and oocyte maturation (Figure 2A,B). Correspondingly, we also found that supplementation of 1 μM MT significantly increased the rates of 2-cell embryos and blastocysts compared to the DT-exposed group (Figure 2C,D) (*p* < 0.05). However, supplementation of 0.1 μM MT did not improve the cleavage and blastocyst rates compared to the DT-exposed group (Figure 2C,D). Based on these observations, we used 1 μM MT for the following experiments. Altogether, these results suggest that the meiotic and developmental failures of DT-exposed oocytes can be rescued by the supplementation of MT.

### 3.3. Melatonin Corrects Chromosome Misalignment and Spindle Disorganization in DT-Exposed Oocytes

It was reported that normal chromosome misalignment and spindle structure is essential for the oocyte nuclear maturation; we thus asked whether DT exposure impairs chromosome alignment and spindle organization. To confirm this possibility, oocytes were stained for α-TUBULIN-FITC, and DAPI was used to display the chromosomes. As shown in Figure 3A, oocytes in control group exhibited linear chromosome morphology and bipolar spindles, whereas the chromosomes were misaligned at the metaphase plate in DT-exposed oocytes, and spindles are also disorganized. We further found that the percentage of oocytes with misaligned chromosomes and disorganized spindles in the DT-exposure group was significantly higher than that in control group (Figure 3B,C) (*p* < 0.05). However, MT supplementation rescued the proportion of oocytes with aberrant chromosome and spindle structure caused by DT exposure (Figure 3B,C). Collectively, these data demonstrate that melatonin protects chromosome and spindle structure in DT-exposed oocytes.

### 3.4. Melatonin Recovers the Assembly of Actin Filaments in DT-Exposed Oocytes

To determine whether DT exposure affects the actin dynamics in oocytes, phalloidin-TRITC staining was performed to label F-actin to observe the polymerization of actin filaments. Fluorescence staining in Figure 4A revealed that oocytes in the control and MT-supplementation group exhibited the uniform distribution of actin filaments with strong signals on the plasma membrane. Conversely, DT-exposed oocytes displayed the altered assembly of actin filaments with weak signals (Figure 4A). In addition, quantitative analysis of fluorescence intensity showed that DT exposure resulted in a substantial reduction in F-action signals compared to the control group (Figure 4B) (*p* < 0.05). MT supplementation significantly increased the F-actin signals in DT-exposed oocytes to the comparable levels in control group (Figure 4B) (*p* < 0.05). Therefore, these results document that melatonin protects the assembly of actin cytoskeleton in DT-exposed oocytes.

### 3.5. Melatonin Restores the Distribution of Mitochondria and Improves the Localization of Cortical Granules in DT-Exposed Oocytes

To explore whether DT exposure impaired the cytoplasmic maturation of oocytes, the distribution of both mitochondria and CGs was marked by Mito Tracker R and PNA-FITC staining, respectively. The fluorescent images in Figure 5A showed that the signals of mitochondria were stable and uniformly distributed around the plasma membrane in control and MT-supplementation group. In contrast, the signals of mitochondria in the DT-exposure group were weak and the specific localization patterns were also disrupted (Figure 5A). Consistently, the quantification analysis also revealed that the fluorescence intensity of mitochondria remarkably decreased in DT-exposed oocytes compared to the control counterparts, but this reduction was rescued by MT supplementation (Figure 5B) (*p* < 0.05). In addition, we observed that the signals of CGs in control group were stable and uniformly distributed under the cortex, DT exposure did not disrupt this specific distribution patterns (Figure 5C). Unexpectedly, the fluorescence intensity of CGs was apparently enhanced in MT-supplemented oocytes relative to the control and DT-exposed counterparts (Figure 5D) (*p* < 0.05). Together, these data imply that melatonin not only rescues the altered distribution of mitochondria of DT-exposed oocytes, but also improves the localization of CGs.

### 3.6. Melatonin Represses Oxidative Stress and DNA Damage of DT-Exposed Oocytes

DT exposure reportedly triggered oxidative stress in several cells to impair cellular functions [26]; we thus attempted to investigate whether DT exposure induced excessive ROS levels in porcine oocytes. The results indicated that ROS signals were very weak in the cytoplasm of control and MT-supplemented oocytes (Figure 6A). However, the signals of ROS in DT-exposed oocytes substantially increased compared to the control and MT group (*p* < 0.05; Figure 6A,B).

Excessive ROS content frequently induces DNA damage [40], which prompted us to detect DNA damage by γH2A.X staining. We observed that DT exposure resulted in a significant increase in the fluorescence intensity of γH2A.X signals in comparison to control group (Figure 6C,D) (*p* < 0.05), indicating a higher frequency of DNA damage in DT-exposed oocytes. By contrast, the fluorescence intensity of γH2A.X signals was remarkably decreased in MT-supplemented oocytes compared to the DT group (Figure 6C,D) (*p* < 0.05). Therefore, these results indicate that melatonin inhibits oxidative stress and DNA damage induced by DT exposure in porcine oocytes.

### 3.7. Melatonin Rescues Excessive Autophagy Induced by Dimethoate Exposure in Porcine Oocytes

It has been shown that excessive oxidative stress always increases autophagy levels in several cells [41]; we thus asked whether DT exposure caused high autophagy activity in porcine oocytes. We found that DT exposure led to a significant increase in protein levels of LC3, an autophagy marker, compared to control group (Figure 7A,B) (*p* < 0.05). Conversely, MT supplementation dramatically decreased the LC3 protein levels in DT-exposed oocytes (Figure 7A,B) (*p* < 0.05), suggesting a reduction in autophagy activity in oocytes. Taken together, these results demonstrate that melatonin rescues autophagy levels in DT-exposed oocytes.

## 4. Discussion

As a widely used pesticide, DT has received increasing attention because of its health risks of humans and animals. Previous studies have shown that DT exposure severely disrupted the functionalities of liver, lung, kidney, and other tissues [11,12,13]. In the reproductive fields, DT exposure reportedly impaired sperm quality and follicle development, in turn causing a low male and female fertility [20,24]. However, the toxic effects of DT exposure on the oocyte maturation and development have not yet been determined.

To address this question, we examined the nuclear and cytoplasmic maturation and developmental competence of porcine oocytes following DT exposure. Our results revealed that DT exposure remarkably reduced the rates of pb1 extrusion and inhibited expansion of cumulus cells, and even DT-exposed oocytes could not develop into blastocysts. These indicated that DT exposure perturbs the normal meiotic maturation and developmental competence of porcine oocytes. It is well-known that the successful completion of meiotic maturation is an essential prerequisite for the acquisition of the developmental capacity of oocytes [42]. In addition, oocyte development is dependent on bidirectional communications between oocytes and cumulus cells [43]. The failure of cumulus expansion likely disrupted the signaling between the two types of cells. It is thus possible that poor developmental competence of DT-exposed porcine oocytes might be the consequences of the impaired meiotic maturation and cumulus expansion.

Numerous studies showed that cytoskeleton organization is tightly correlated with the meiotic maturation of oocytes [44]. The establishment of a bipolar spindle is required for chromosome segregation and cell division in germ cells [45]. The polymerization of actin filament participates in modulating the localization and movement of organelles in cells [46]. Herein, we found that DT exposure severely impaired the assembly of bipolar spindle structure, chromosome alignment, and actin polymerization of porcine oocytes. Therefore, the failure of meiotic maturation of DT-exposed porcine oocytes could be attributed to the dysfunctional organization of cytoskeleton. Furthermore, the correct distribution of both mitochondria and CGs has been considered as two important indicators of oocyte cytoplasmic maturation [47]. The mitochondria primarily generate energy essential for oocyte maturation process [48]. Our data showed that the distribution of mitochondria was disrupted in oocytes exposed to DT, suggesting that DT exposure prevented cytoplasmic maturation of porcine oocytes. The translocation of CGs during meiotic maturation and proteins stored in CGs are necessary for oocyte cytoplasmic maturation [49]. Unexpectedly, CGs did not display the aberrant distribution in DT-exposed oocytes, although a normal distribution of CGs is required to block polyspermy [50]. This might be due to a high resistance of CG components in porcine oocytes to DT exposure. A number of studies have shown that environmental pollutant exposure generally activates the process of oxidative stress in cells [51]. Concomitantly, the increased oxidative stress induces DNA damage and excessive autophagy, thereby impairing cellular functions [40,41]. In this study, we observed the increased ROS levels, DNA damage accumulation, and excessive autophagy in porcine oocytes exposed DT. Hence, oxidative-stress-induced DNA damage and abnormal autophagy could account for the poor meiotic maturation of DT-exposed porcine oocytes.

Recent studies have revealed that natural and artificial antioxidants are commonly able to overcome the toxic effects of environmental pollutants in cells [27]. Our results showed that MT could rescue the meiotic and developmental defects of DT-exposed oocytes through reducing the excessive ROS, γH2A.X, and autophagy levels. Consistent with our data, previous studies showed that MT alleviates ROS accumulation to protect oocyte maturation in several species [32,36]. In addition, excessive autophagy impaired cell viability [52]. MT was previously reported to antagonize autophagy activity under other cellular contexts [53]. Thus, we speculated that MT could reduce excessive autophagy induced by DT to relieve its toxicity in porcine oocytes. Indeed, emerging evidence has indicated that MT acts as a strong antioxidant to relieve environmental pollutant-induced impairments of oocyte maturation and early embryonic development in several species [36,37,38]. These data revealed a conserved role of MT in antagonizing the toxicity of environmental chemicals in both oocyte maturation and acquisition of developmental competence between species.

## 5. Conclusions

These results demonstrated that DT exposure caused meiotic and developmental defects of porcine oocytes. Melatonin administration counteracted the deleterious effects of DT exposure by preventing the oxidative stress. These findings imply that melatonin could be a promising agent in improving the quality of DT-exposed oocytes from humans and animals.

## Figures and Tables

**Figure 1 animals-12-00832-f001:**
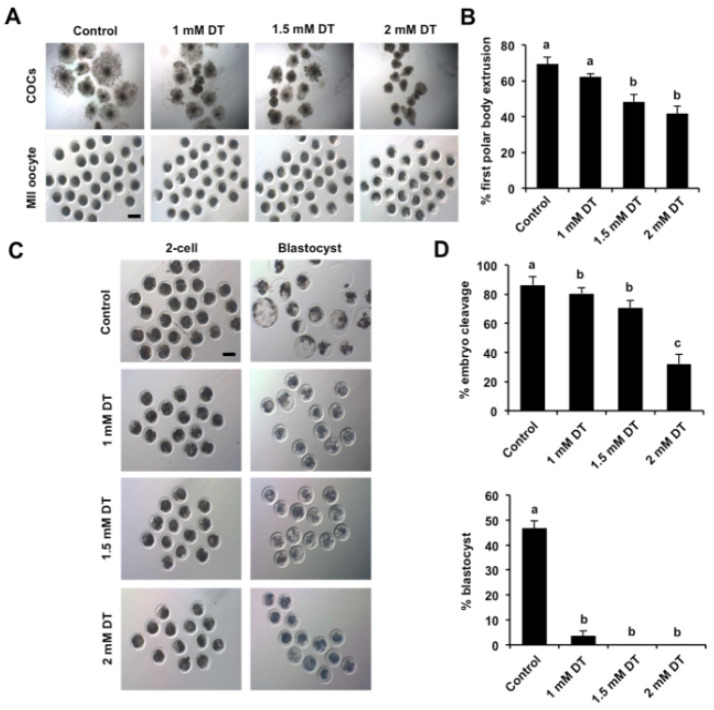
Effects of different concentrations of DT on porcine oocyte maturation. (**A**) Representative images of cumulus cell-enclosed and denuded oocytes in control and different concentrations of DT-exposed groups. The experiment was independently repeated six times with at least 140 oocytes per group. MII, metaphase II. Scale bar: 100 µm. (**B**) The rates of first polar body extrusion in the control and DT-exposed groups. The number of oocytes with first polar body after in vitro maturation for 44 h was recorded and the rates of first polar body extrusion were statistically analyzed. (**C**) Representative images of 2-cell embryos and blastocysts in control and DT-exposed groups. Scale bar: 100 µm. (**D**) The developmental rates of early embryos derived from parthenogenetically activated oocytes. The rates of 2-cell embryos and blastocysts were statistically analyzed in the control and DT-exposed groups. All data are shown as mean ± S.E.M and different letters on the bars indicate significant differences (*p* < 0.05).

**Figure 2 animals-12-00832-f002:**
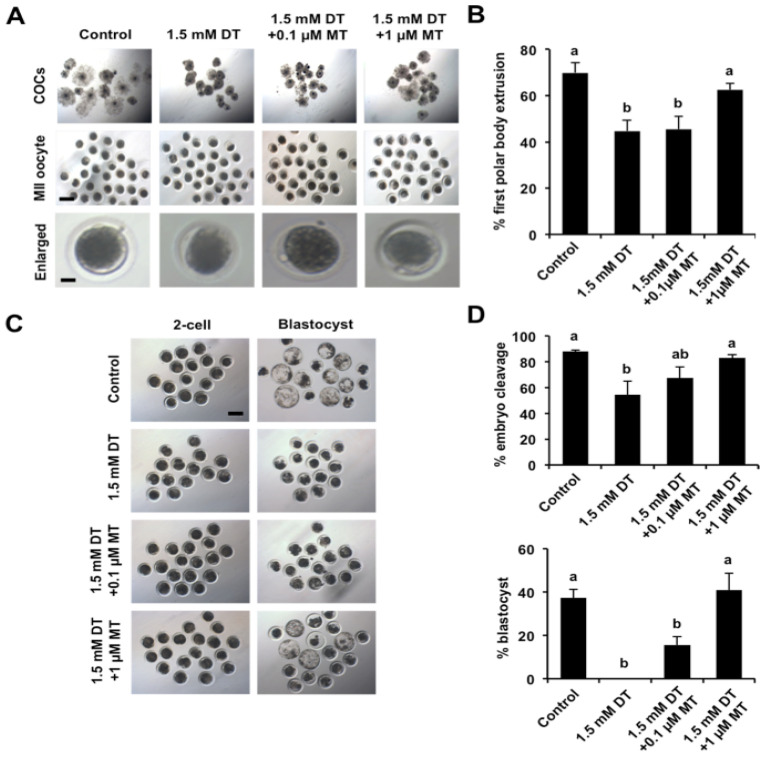
Effects of different concentrations of DT and MT on porcine oocyte maturation. (**A**) Representative images of cumulus cell-enclosed and denuded oocytes in each group. The experiment was independently repeated six times with at least 120 oocytes per group.MII, metaphase II. Scale bar: 100 µm. (**B**) The rates of first polar body extrusion in control and MT-supplemented groups with 1.5 mM DT. (**C**) Representative images of 2-cell embryos and blastocysts in control and MT-supplemented groups with 1.5 mM DT. Scale bar: 100 µm. (**D**) The developmental rate of early embryos derived from parthenogenetically activated oocytes. The rates of 2-cell embryos and blastocysts were recorded and statistically analyzed in each group. All data are shown as mean ± S.E.M and different letters on the bars indicate significant differences (*p* < 0.05).

**Figure 3 animals-12-00832-f003:**
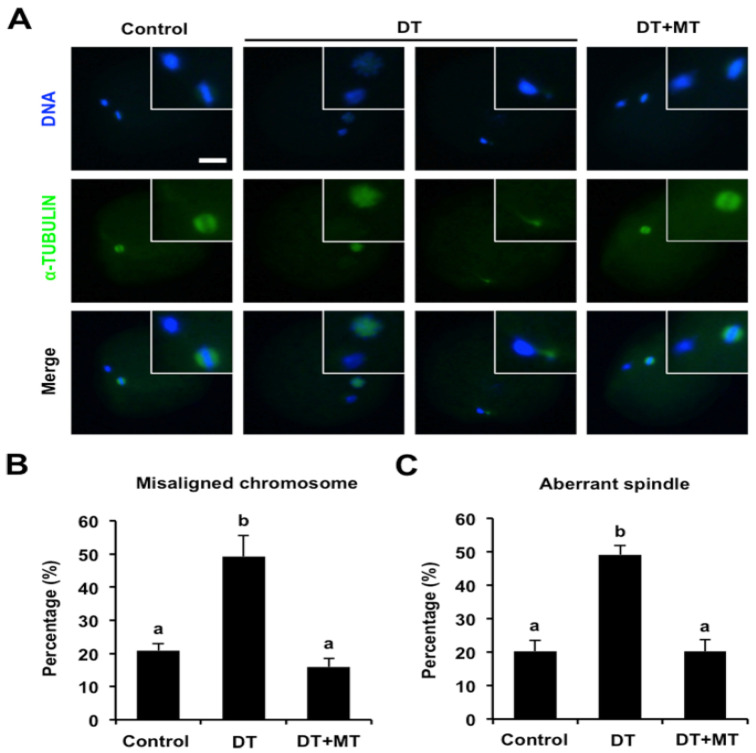
Effects of MT supplementation on chromosome and spindle defects in DT-exposed porcine oocytes. (**A**) Representative images of chromosome and spindle morphology. Oocytes in control, DT-exposed and MT-supplemented groups were stained for α-TUBULIN (green) and DNA (blue). The experiment was independently repeated three times with at least 40 oocytes per group. Bottom panel shows the merged images between α-TUBULIN and DNA. White square insets indicate chromosomes and spindles at high magnification. Scale bar: 50 µm. (**B**) The rates of misaligned chromosomes were recorded and statistically analyzed in control, DT-exposed and MT-supplemented groups. (**C**) The rates of aberrant spindles were recorded and statistically analyzed in control, DT-exposed and MT-supplemented groups. All data are presented as mean ± S.E.M and different letters on the bars indicate significant differences (*p* < 0.05).

**Figure 4 animals-12-00832-f004:**
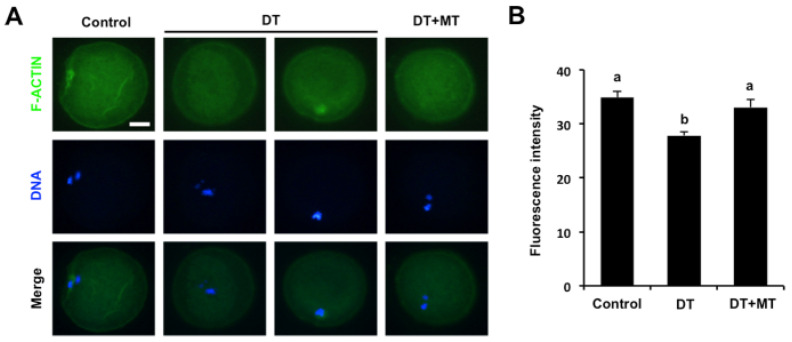
Effects of MT supplementation on the actin polymerization in DT-exposed porcine oocytes. (**A**) Representative images of actin filaments in control, DT-exposed and MT-supplemented groups. Oocytes in each group were stained for phalloidin (green) and DNA (blue). The experiment was independently repeated three times with at least 30 oocytes per group. Scale bar: 50 µm. (**B**) The fluorescence intensity of actin signals was measured and statistically analyzed in control, DT-exposed and MT-supplemented groups. All data are shown as mean ± S.E.M and different letters on the bars indicate significant differences (*p* < 0.05).

**Figure 5 animals-12-00832-f005:**
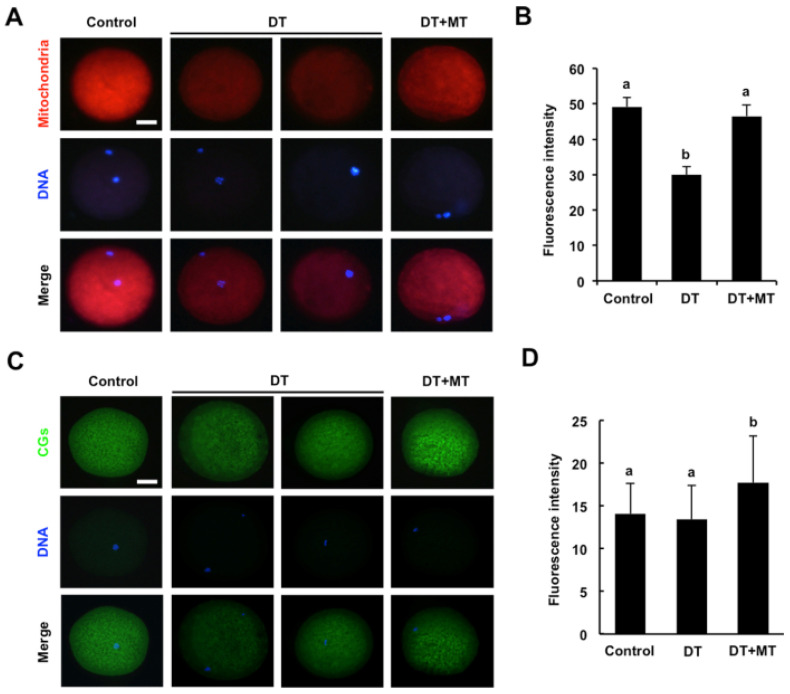
Effects of MT supplementation on the distribution of mitochondria and cortical granules in DT-exposed porcine oocytes. (**A**) Representative images of mitochondria in control, DT-exposed and MT-supplemented groups. Oocytes in each group were stained for MitoTracker (red) and DNA (blue). The experiment was independently repeated three times with at least 35 oocytes per group. Scale bar: 50 µm. (**B**) The fluorescence intensity of mitochondrion signals was measured in control, DT-exposed and MT-supplemented groups. (**C**) Representative images of cortical granules localization in control, DT-exposed and MT-supplemented groups. The experiment was independently repeated three times with at least 40 oocytes per group. Scale bar: 50 µm. (**D**) The fluorescence intensity of cortical granules was measured in control, DT-exposed and MT-supplemented groups. All data are presented as mean ± S.E.M and different letters on the bars indicate significant differences (*p* < 0.05).

**Figure 6 animals-12-00832-f006:**
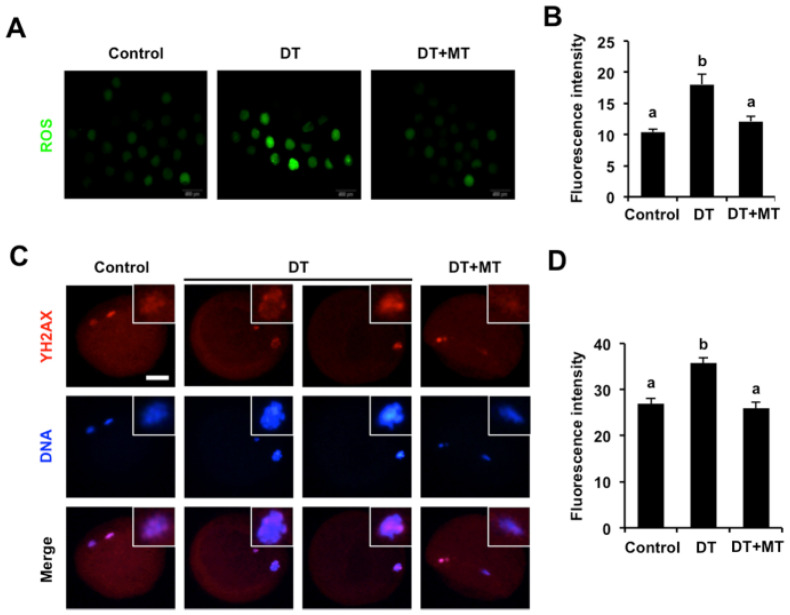
Effects of MT supplementation on ROS levels and DNA damage in DT-exposed porcine oocytes. (**A**) Representative images of ROS staining in control, DT-exposed and MT-supplemented groups. Oocytes in each group were stained for DCFH-DA (green) and DNA (blue). The experiment was independently repeated three times with at least 45 oocytes per group. Scale bar: 50 µm. (**B**) The fluorescence intensity of ROS was analyzed statistically in control, DT-exposed and MT-supplemented groups. (**C**) Representative images of DNA damage in control, DT-exposed and MT-supplemented groups. The experiment was independently repeated three times with at least 30 oocytes per group. Scale bar: 50 µm. (**D**) The fluorescence intensity of γH2A.X signals was measured in control, DT-exposed and MT-supplemented groups. All data are presented as mean ± S.E.M and different letters on the bars indicate significant differences (*p* < 0.05).

**Figure 7 animals-12-00832-f007:**
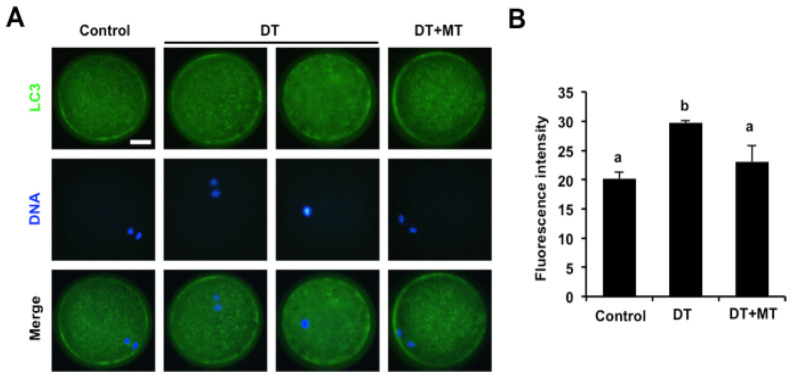
Effects of MT supplementation on autophagy activity in DT-exposed porcine oocytes. (**A**) Representative images of LC3 staining in control, DT-exposed and MT-supplemented groups. Oocytes in each group were stained for LC3 (green) and DNA (blue). The experiment was independently repeated three times with at least 30 oocytes per group. Scale bar: 50 µm. (**B**) The fluorescence intensity of LC3 was recorded in control, DT-exposed and MT-supplemented groups. All data are shown as mean ± S.E.M and different letters on the bars indicate significant differences (*p* < 0.05).

## Data Availability

The data presented in this study are available on request from the corresponding author.

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
