# Peer review of "Melatonin Rescues Dimethoate Exposure-Induced Meiotic and Developmental Defects of Porcine Oocytes"

_animals, 2022, doi:10.3390/ani12070832_

Round 1
Reviewer 1 Report
General comment
The study aimed to assess the toxic effects of dimethoate exposure on meiotic maturation and how melatonin incorporation alleviates such effects. The authors used the porcine model in this study. In general, the manuscript fits the scope of the journal. Some comments should be clarified well, and I suggest a moderate revision based on the appended comments:
Line 79: The authors should improve the rationale of the study
Line 79: why did the authors select the porcine oocytes to be the model in this study?
Line 86: the authors should indicate the company source of the materials used in the study.
Line 86: Experimental design figure should be incorporated in the manuscript.
Line 86: group assignment should be clarified.
Line 94: Antral follicles sizes???
Line 100-142: please add the relevant references to each part of the methodology.
Line 185: In the results: P-value should be indicated in all figures.
Line 395: this figure is not clear. Please improve it.
Line 140: which test was used for the statistics.
Line 142: The post-hoc test was not used in this study.
Line 401: In the discussion section, the authors should explain different pathways that melatonin rescues the negative effect of dimethoate on porcine oocyte developmental competence and how they verified the results.
Author Response
The study aimed to assess the toxic effects of dimethoate exposure on meiotic maturation and how melatonin incorporation alleviates such effects. The authors used the porcine model in this study. In general, the manuscript fits the scope of the journal. Some comments should be clarified well, and I suggest a moderate revision based on the appended comments.
Response: Thank you for the good comments from the reviewer. We clearly have made a point-to-point response to all of questions and concerns raised by the reviewer.
Line 79: The authors should improve the rationale of the study.
Response: Thank you for the good comment. We have added some explanations and cited a reference to justify the usage of porcine oocytes in the revised manuscript.
Line 79: why did the authors select the porcine oocytes to be the model in this study?
Response: Thank you for the good comment. Given the size and maturation time of porcine oocytes are more similar to humans compared to mice, porcine oocytes are thus used to be the model in this study. We have added some sentences to explain the rationale of usage of porcine oocytes in the revised manuscript.
Line 86: the authors should indicate the company source of the materials used in the study.
Response: Thank you for the good suggestion. The majority of reagents in this study are purchased from Sigma company. The reagents that have been not marked with company name should be purchased from Sigma. Chemicals from other companies have been marked with specific vendor name in the supplementary Table S1. Finally, we have added an instruction for company source of materials in the revised manuscript. Namely, all reagents used in this study were purchased from Sigma (Sigma-Aldrich, St Lou-is, MO) unless otherwise stated.
Line 86: Experimental design figure should be incorporated in the manuscript.
Response: Thank you for the good comment. The presentation of results in the manuscript can embody the thread of experimental designs. Specifically, screening of dimethoate and melatonin concentrations, assessment of cytoplasmic and nuclear maturation, analyses of reactive oxygen species and autophagy, and DNA damage.
Line 86: group assignment should be clarified.
Response: Thank you for the good comment. The assignment of experimental groups in this study has been clearly displayed in each Figure.
Line 94: Antral follicles sizes???
Response: Thank you for the good suggestion. We have added the sizes of antral follicles in our revised manuscript.
Line 100-142: please add the relevant references to each part of the methodology.
Response: Thank you for the good comment. Although most of common reagents were used to carry out the methods in the manuscript, the specific procedures of each method used in this study differ from those reported in other literatures. In addition, wordings of each method used in the manuscript are different from other articles. Therefore, we did not insert any references to each part of methodology in the manuscript.
Line 185: In the results: P-value should be indicated in all figures.
Response: Thank you for the good suggestion. P-value has been indicated in the main texts when results were described. Different letters among groups were used to denote statistical significance in Figures.
Line 395: this figure is not clear. Please improve it.
Response: Thank you for the good comment. We tried to find high quality images to replace the present figures, but they are the best ones in the acquired images. Overall, the present images did not allow readers to misunderstand its meaning.
Line 140: which test was used for the statistics.
Response: Thank you for the good comment. We have indicated in the main text that one-way ANOVA test was used to perform statistical analyses for data among three groups. Student’s t test was used to analyze the statistical significance for data between two groups.
Line 142: The post-hoc test was not used in this study.
Response: Thank you for the good reminder. SPSS software in this study was run to perform one-way ANOVA and Student’s t test. At the same time, post-hoc test was also performed in the last step of SPSS analysis. Thus, we could obtain statistical significance between any two groups. As shown in all figures, different letters on the bars represent statistical significance, while same letters on the bars indicate no statistical significance.
Line 401: In the discussion section, the authors should explain different pathways that melatonin rescues the negative effect of dimethoate on porcine oocyte developmental competence and how they verified the results.
Response: Thank you for the good comments. We have added more discussions to explain different pathways that melatonin rescues the negative effect of dimethoate on porcine oocyte maturation. The specific information was provided as following. Consistent with our data, previous studies showed that MT alleviates ROS accumulation to protect oocyte maturation in several species. In addition, excessive autophagy impaired cell viability. MT was previously reported to antagonize autophagy activity under other cellular contexts. Thus, we speculated that MT could reduce excessive autophagy induced by DT to relieve its toxicity in porcine oocytes.
Reviewer 2 Report
Authors examine the effect of melatonin for porcine oocyte maturation. Experimental design is well described and simple.
And I wonder why authors treat the melatonin and methoate during overall in vitro maturation full period.
Author Response
Authors examine the effect of melatonin for porcine oocyte maturation. Experimental design is well described and simple.
Response: Thank you for the good comment from the reviewer. We have made a point-to-point response to all of questions raised by the reviewer.
And I wonder why authors treat the melatonin and dimethoate during overall in vitro maturation full period.
Response: Thank you for the good comment. In this study, we aim to investigate whether dimethoate exposure affects meiotic and developmental potential of porcine oocytes. Meanwhile, whether melatonin can overcome the toxic effects of dimethoate on meiotic maturation and development of porcine oocytes should be explored. In addition, a previous study has shown that dimethoate exposure blocked the development of antral follicles in mice. The data give us some implications that dimethoate probably affects the whole maturation period of oocytes. Hence, porcine oocytes were exposed to dimethoate during overall in vitro maturation period.
Reviewer 3 Report
Review of Manuscript ID: animals-1617318
Melatonin rescues dimethoate exposure-induced meiotic and developmental defects of porcine oocytes
Authors: Qi Jiang, Chi Ding, Xingyu Liu, Yuanyuan Lei, Siying Li and Zubing Cao
Key Findings: This study showed the toxicity of dimethoate (DT) on porcine oocyte maturation and the protective effects of melatonin (MT) on DT-exposed oocytes. DT exposure with 1.5 mM partially inhibited cumulus cell expansion and significantly reduced the rate of first polar body extrusion (pb1) during oocyte maturation. Parthenogenetically activated embryos derived from DT-exposed oocytes could not develop to the 2-cell and blastocyst stage. Furthermore, DT exposure led to a significant increase in the rates of both misaligned chromosomes and disorganized spindles and abnormal actin assembly. Meanwhile, DT exposure severely disrupted the distribution patterns of mitochondria in oocytes but did not change the subcellular localization of cortical granules. Significantly, MT supplementation rescued the meiotic and developmental defects of DT-exposed oocytes through repressing the generation of excessive reactive oxygen species (ROS) and autophagy and DNA damage accumulation. These results demonstrate that melatonin protects against meiotic defects induced by DT during porcine oocyte maturation.
Experimental Rationale: As a widely used pesticide, DT has received increasing attention because of its health risks to humans and animals. Previous studies have shown that DT exposure severely disrupted the functionalities of the liver, lung, kidney, and other tissues. DT exposure also has been reported to impair sperm quality and follicle development, thus causing low male and female fertility. However, the toxic effects of DT exposure on oocyte maturation and development have not yet been determined.
The authors examined the nuclear and cytoplasmic maturation and developmental competence of porcine oocytes following DT exposure to address this question. Their results revealed that DT exposure remarkably reduced the rates of pb1 extrusion and inhibited the expansion of cumulus cells, and even DT-exposed oocytes could not develop into blastocysts. These findings indicated that DT exposure perturbs porcine oocytes' normal meiotic maturation and developmental competence. It is well-known that the successful completion of meiotic maturation is an essential prerequisite for acquiring the developmental capacity of oocytes. In addition, oocyte development is dependent on bidirectional communications between oocytes and cumulus cells. The failure of cumulus expansion likely disrupted the signaling between the two types of cells. It is thus possible that poor developmental competence of DT-exposed porcine oocytes might be the consequence of the impaired meiotic maturation and cumulus expansion.
Experimental Design and Methodology: The experimental design and methodology appear to be sound, complete, and appropriate. Sufficient details and references are reported to aid future investigators. The investigators followed experimental procedures that have been reported in previous studies, including earlier studies by the current investigators. Statistical analyses seemed appropriate, but relatively few statistical details were provided.
Results: The results are presented in a straightforward and clear manner in tables, figures, and images. The results presented are appropriate for evaluating the stated research objectives.
Conclusions: Points raised in the discussion and the conclusions are appropriate based upon the results presented. Their conclusion that DT exposure caused meiotic and developmental defects of porcine oocytes and that melatonin administration counteracted the deleterious effects of DT exposure by preventing oxidative stress are logical and supported by their results.
Language and Editorial Improvements: The manuscript was well written and easy to read. Some very minor editorial corrections may be in order. This reviewer did not address these changes.
Recommendation: This manuscript does provide new, potentially important, and unique information to the literature regarding the mechanisms by which DT exposure causes meiotic and developmental defects of porcine oocytes through disrupting the chromosome and spindle organization, actin filament assembly, mitochondria, and cortical granule distribution. These findings indicate that melatonin could improve the quality of DT-exposed oocytes from animals and humans. I recommend acceptance and publication following minor editorial revision.
Author Response
Review of Manuscript ID: animals-1617318
Melatonin rescues dimethoate exposure-induced meiotic and developmental defects of porcine oocytes
Authors: Qi Jiang, Chi Ding, Xingyu Liu, Yuanyuan Lei, Siying Li and Zubing Cao
Key Findings: This study showed the toxicity of dimethoate (DT) on porcine oocyte maturation and the protective effects of melatonin (MT) on DT-exposed oocytes. DT exposure with 1.5 mM partially inhibited cumulus cell expansion and significantly reduced the rate of first polar body extrusion (pb1) during oocyte maturation. Parthenogenetically activated embryos derived from DT-exposed oocytes could not develop to the 2-cell and blastocyst stage. Furthermore, DT exposure led to a significant increase in the rates of both misaligned chromosomes and disorganized spindles and abnormal actin assembly. Meanwhile, DT exposure severely disrupted the distribution patterns of mitochondria in oocytes but did not change the subcellular localization of cortical granules. Significantly, MT supplementation rescued the meiotic and developmental defects of DT-exposed oocytes through repressing the generation of excessive reactive oxygen species (ROS) and autophagy and DNA damage accumulation. These results demonstrate that melatonin protects against meiotic defects induced by DT during porcine oocyte maturation.
Response: Thank you for the good comments from the reviewer. We have made a point-to-point response to all of questions and concerns raised by the reviewer.
Experimental Rationale: As a widely used pesticide, DT has received increasing attention because of its health risks to humans and animals. Previous studies have shown that DT exposure severely disrupted the functionalities of the liver, lung, kidney, and other tissues. DT exposure also has been reported to impair sperm quality and follicle development, thus causing low male and female fertility. However, the toxic effects of DT exposure on oocyte maturation and development have not yet been determined.
Response: Many thanks for the good comments.
The authors examined the nuclear and cytoplasmic maturation and developmental competence of porcine oocytes following DT exposure to address this question. Their results revealed that DT exposure remarkably reduced the rates of pb1 extrusion and inhibited the expansion of cumulus cells, and even DT-exposed oocytes could not develop into blastocysts. These findings indicated that DT exposure perturbs porcine oocytes' normal meiotic maturation and developmental competence. It is well-known that the successful completion of meiotic maturation is an essential prerequisite for acquiring the developmental capacity of oocytes. In addition, oocyte development is dependent on bidirectional communications between oocytes and cumulus cells. The failure of cumulus expansion likely disrupted the signaling between the two types of cells. It is thus possible that poor developmental competence of DT-exposed porcine oocytes might be the consequence of the impaired meiotic maturation and cumulus expansion.
Response: Thank you for the good comments.
Experimental Design and Methodology: The experimental design and methodology appear to be sound, complete, and appropriate. Sufficient details and references are reported to aid future investigators. The investigators followed experimental procedures that have been reported in previous studies, including earlier studies by the current investigators. Statistical analyses seemed appropriate, but relatively few statistical details were provided.
Response: Thank you for the good comments. We have added more statistical details in the revised manuscript.
Results: The results are presented in a straightforward and clear manner in tables, figures, and images. The results presented are appropriate for evaluating the stated research objectives.
Response: Thank you for the good comments.
Conclusions: Points raised in the discussion and the conclusions are appropriate based upon the results presented. Their conclusion that DT exposure caused meiotic and developmental defects of porcine oocytes and that melatonin administration counteracted the deleterious effects of DT exposure by preventing oxidative stress are logical and supported by their results.
Response: Thank you for the good comments.
Language and Editorial Improvements: The manuscript was well written and easy to read. Some very minor editorial corrections may be in order. This reviewer did not address these changes.
Response: Thank you for the good comments. We have corrected some errors at the aspects of grammars, spelling, and wordings in the revised manuscript.
Recommendation: This manuscript does provide new, potentially important, and unique information to the literature regarding the mechanisms by which DT exposure causes meiotic and developmental defects of porcine oocytes through disrupting the chromosome and spindle organization, actin filament assembly, mitochondria, and cortical granule distribution. These findings indicate that melatonin could improve the quality of DT-exposed oocytes from animals and humans. I recommend acceptance and publication following minor editorial revision.
Response: Thank you for the good comments.
Reviewer 4 Report
This manuscript shows interesting results for the scientific community about the impact of dimethoate on porcine oocytes maturation and how supplementation with melatonin can restore the developmental competence of oocytes in this species.
However, the manuscript still requires major improvements before publication.
L24 ‘’ to counteract it have not yet been known ‘’, is not well written. The authors should replace this sentence by ‘’ to counteract it are not yet known’’.
In L24, the next sentence: ‘’Here, we show that the toxicity of DT on porcine oocyte 24 maturation and the protective effects of melatonin (MT) on DT-exposed oocytes’’ is a little bit confusing for readers, so I recommend changing the verb ‘’show that’’ by ‘’we studied or investigated….’’
In L29, the authors should remove ‘’of both’’ if they add to the sentence ‘’and abnormal actin assembly’’.
In L30 ‘’meanwhile’’ is not properly used.
In L54, it would be interesting if the authors specified what kind of reproductive defects are produced in male mice.
L62-64, please rewrite this sentence for a better understanding.
L75-76, please rewrite this sentence for a better understanding.
MATERIAL AND METHODS
In subsection ‘’ 2.1 Preparation of dimethoate and melatonin’’, the authors should clarify the composition of maturation medium used, the working solutions of DT and MT, and the initial concentration of DMSO to prepare the stock solutions.
The authors should specify that the oocytes are activated by parthenogenesis and explain how long the embryos are cultured as well as when the embryo cleavage rates and blastocysts are recorded in material and methods, subsection 2.3 Oocyte activation and in vitro culture.
There are two 2.3 subsections, please correct it.
The authors should add the oocyte’s target proteins used in this study at the beginning of the subsection 2.3 Immunofluorescence staining. and briefly explain why they selected such proteins and their function. Perhaps this subsection can be divided and describe it separately each protocol.
Information on reagents used in this study should be removed from Table S1, since it is not mentioned in any part of the manuscript, I suggest adding the information of vendor in the manuscript. In addition, primary and secondary antibody information must be unified in the same table.
Please, specify the number of oocytes evaluated in each analysis and in each treatment (control or DT or MT).
When do you used one-way ANOVA or student’s t test? Do you check if the data meet a normal distribution first?
RESULTS
This section should only explain those findings obtained after the statistical analysis. The introductory sentence of every subsection seems more appropriate for material and methods and not for this section, while the last sentence seems more appropriate for discussion. I recommend eliminating those parts from results and placing them in the mentioned sections.
L152, please add ‘’compared to control group’’.
L155, I suggest emphasizing in the following sentence that the lowest cleavage rate was obtained after exposing oocytes to 2 mM of DT, since this data it’s interesting.
L157, please add ‘’compared to control group’’. Replace but by and.
Figure legend 1 should be shortened. These sentences can be removed ’’GV oocytes were matured in vitro for 44 h in the presence of the indicated concentration of DT. Oocytes were cultured in maturation medium containing equivalent amount of DMSO and were served as the control group. The experiment was independently repeated six times with at least 140 oocytes 190 per group.’’ In fact, the number of oocytes should be added to material and method section and not here.
In L190, the authors said ‘’ The experiment was independently repeated six times’’, however, in the statistical analysis they commented that the experiment was replicated three times only. Is this a mistake?
L201-205, should be removed. L204-205 can be added to material and methods.
L207, please check the concentration of MT, I assume that authors wanted to say 0.1µM MT.
Perhaps, it would be interesting to emphasize in subsection 3.2, that MT restored the percentage of pb1 extrusion, embryo cleavage and blastocysts to similar levels to those observed in the control group.
Figure legend 2 should be shortened. L239 and L245 can be removed.
In section 3.3, L254-256 and then L256-258 sounds repetitive. Perhaps the authors could unify both sentences and reorganize figure 3. The significant differences in the proportion of misaligned chromosomes and aberrant spindles between control, DT and DT+MT groups can be described first (fig 3B and 3C) and then the authors can say that representative images about the changes in chromosome and spindle organization can be observed in figure 3A (to support the results of fig 3B and 3C).
I didn’t find the protocol of polymerization of actin filaments in material and methods, this part should be added. L293-294 correspond again to mat and methods.
L314-315 should be removed.
L357-359 should be removed or added to discussion.
L360-361, please add compared to the control and MT group (P < 0.05; Fig. 6A and 6B) and removed L361-364 to avoid repetition. By adding this, you can also shorten this paragraph.
L365 should be added to material and methods and explain better the subsection 2.3 of immunostaining. Perhaps, this subsection of material and methods can be divided and explain separately the protocol of evaluating chromosome and spindle organization, actin polymerization, DNA damage and autophagy.
Why do you use γH2A.X for detecting DNA damage instead of the TUNEL assay?
L370, please add compared to the DT group.
L385-386 should be removed or added to discussion.
L387-388 should be removed or added to materials and methods.
L389, please add ‘’ We found that DT exposure led to a significant increase in protein levels of LC3, an autophagy marker, compared to control group.
DISCUSSION
In general, this section needs improvements. A better connection of all the results obtained in the study are necessary.
I suggest connecting all the results that can be responsible for the poor meiotic maturation by the end of the discussion and emphasize how the lower meiotic and cytoplasmatic maturation of DT oocytes as well as their lower developmental competence affects the embryo and blastocyst rate.
I missed some comparisons of the results obtained in oocyte maturation and its developmental competence with similar papers using another environmental pollutant, specially about the role of melatonin.
L402-407. This first paragraph is quite similar to the introduction. Therefore, it sounds repetitive and can be avoided.
L428, what is the role of CGs in oocyte cytoplasmic maturation?
L432-433 should be rewritten, I imagine that you are saying that a normal distribution of CGs is required to block polyspermy, but it is not well written.
In the last paragraph, it would be interesting to highlight the beneficial effects of MT on embryo cleavage rate and blastocysts, which also proves that MT restores oocyte maturation and their developmental competence.
Author Response
This manuscript shows interesting results for the scientific community about the impact of dimethoate on porcine oocytes maturation and how supplementation with melatonin can restore the developmental competence of oocytes in this species. However, the manuscript still requires major improvements before publication.
Response: Thank you for the good comments from the reviewer. We have made a point-to-point response to all of questions and concerns raised by the reviewer.
L24 ‘’ to counteract it have not yet been known ‘’, is not well written. The authors should replace this sentence by ‘’ to counteract it are not yet known’’.
Response: Thank you for the good suggestion. We have replaced “to counteract it have not yet been known” by “to counteract it are not yet known” in the revised manuscript.
In L24, the next sentence: ‘’Here, we show that the toxicity of DT on porcine oocyte 24 maturation and the protective effects of melatonin (MT) on DT-exposed oocytes’’ is a little bit confusing for readers, so I recommend changing the verb ‘’show that’’ by ‘’we studied or investigated….’’
Response: Thank you for the good suggestion. We have replaced “show” by “investigated” in the revised manuscript.
In L29, the authors should remove ‘’of both’’ if they add to the sentence ‘’and abnormal actin assembly’’.
Response: Thank you for the good comment. We have removed “of both” in the revised manuscript.
In L30 ‘’meanwhile’’ is not properly used.
Response: Thank you for the good comment. We have deleted “meanwhile” in the revised manuscript.
In L54, it would be interesting if the authors specified what kind of reproductive defects are produced in male mice.
Response: Thank you for the good comment. We have added the specific information on reproductive defects in the revised manuscript.
L62-64, please rewrite this sentence for a better understanding.
Response: Thank you for the good comment. We have reworded the sentence in the revised manuscript. The specific description is indicated as follows. Moreover, maternal exposure to DT not only reduced the number of both implantations and live fetuses, but also increased the incidences of early embryo resorption.
L75-76, please rewrite this sentence for a better understanding.
Response: Thank you for the good comment. We have reworded the sentence in the revised manuscript.
MATERIAL AND METHODS
In subsection ‘’ 2.1 Preparation of dimethoate and melatonin’’, the authors should clarify the composition of maturation medium used, the working solutions of DT and MT, and the initial concentration of DMSO to prepare the stock solutions.
Response: Thank you for the good comments. We have specified the composition of maturation medium in the 2.2 subsection in the revised manuscript. The specific description is indicated as follows. The working solutions of DT and MT have been indicated in the first and second section of results. The initial concentration of DMSO did not affect oocyte maturation and the final concentration of DMSO is associated with oocyte maturation.
The authors should specify that the oocytes are activated by parthenogenesis and explain how long the embryos are cultured as well as when the embryo cleavage rates and blastocysts are recorded in material and methods, subsection 2.3 Oocyte activation and in vitro culture.
Response: Thank you for the good comments. We have specified the content in the revised manuscript. The specific description is indicated as follows. Embryos were cultured for 7 days at 38.5 °C, 5% CO2 and 95% air with saturated humidity. The number of cleaved embryos and blastocysts were recorded at day 2 and day 7, respectively.
There are two 2.3 subsections, please correct it.
Response: Thank you for the nice reminder. We have corrected the number order in the revised manuscript.
The authors should add the oocyte’s target proteins used in this study at the beginning of the subsection 2.3 Immunofluorescence staining. And briefly explain why they selected such proteins and their function. Perhaps this subsection can be divided and describe it separately each protocol.
Response: Thank you for the good comments. Given the description of oocyte’s target proteins would occupy the space in the main text, we have specified these target proteins in the section of results and Table S1 in the revised manuscript. In addition, the procedure of immunofluorescence staining for each target protein is similar. Hence, the method of immunofluorescence staining should be placed together.
Information on reagents used in this study should be removed from Table S1, since it is not mentioned in any part of the manuscript, I suggest adding the information of vendor in the manuscript. In addition, primary and secondary antibody information must be unified in the same table.
Response: Thank you for the good comments. Information on reagents listed in Table S1 has been indicated in the sections of “Determination of mitochondrial distribution”, “Evaluation of cortical granules distribution”, and “Determination of ROS levels”. The information should be kept to help readers repeat the experiments. The primary and secondary antibody information has been unified in the Table S1.
Please, specify the number of oocytes evaluated in each analysis and in each treatment (control or DT or MT).
Response: Thank you for the good suggestion. We have specified the number of oocytes evaluated in each experiment in figure legends.
When do you used one-way ANOVA or student’s t test? Do you check if the data meet a normal distribution first?
Response: Thank you for the good comment. We used one-way ANOVA for statistical analysis when the data were compared among three groups. Additionally, the student’s t test was performed to compare the significant difference of the data between two groups. We have confirmed that the data met a normal distribution before the onset of statistical analyses.
RESULTS
This section should only explain those findings obtained after the statistical analysis. The introductory sentence of every subsection seems more appropriate for material and methods and not for this section, while the last sentence seems more appropriate for discussion. I recommend eliminating those parts from results and placing them in the mentioned sections.
Response: Thank you for the good comments. The introductory sentence of every subsection can help readers understand the aim of the experiments. The last sentence of every section is a short summary of results, which help readers capture the main findings of each section. Therefore, the introductory and summary sentence of each section can make the manuscript more readable and logic.
L152, please add ‘’compared to control group’’.
Response: Thank you for the good suggestion. We have added “compared to control group” in the revised manuscript as suggested.
L155, I suggest emphasizing in the following sentence that the lowest cleavage rate was obtained after exposing oocytes to 2 mM of DT, since this data it’s interesting.
Response: Thank you for the good suggestion. We have added “compared to control group” in the revised manuscript as suggested.
L157, please add ‘’compared to control group’’. Replace but by and.
Response: Thank you for the good suggestion. We have added “compared to control group” and have replaced “but” by “and” in the revised manuscript.
Figure legend 1 should be shortened. These sentences can be removed ’’GV oocytes were matured in vitro for 44 h in the presence of the indicated concentration of DT. Oocytes were cultured in maturation medium containing equivalent amount of DMSO and were served as the control group. The experiment was independently repeated six times with at least 140 oocytes 190 per group.’’ In fact, the number of oocytes should be added to material and method section and not here.
Response: Thank you for the good comments. We have removed these sentences in the revised manuscript as suggested. In addition, we have added the number of oocytes used in each experiment in figure legends.
In L190, the authors said ‘’ The experiment was independently repeated six times’’, however, in the statistical analysis they commented that the experiment was replicated three times only. Is this a mistake?
Response: Thank you for the good comments. In fact, we commented in the statistical analysis that three replicates were at least performed for each experiment. Therefore, the description on the statistical analysis is correct in the manuscript.
L201-205, should be removed. L204-205 can be added to material and methods.
Response: Thank you for the good comments. The introductory sentence of every subsection can help readers understand the aim of the experiments.
L207, please check the concentration of MT, I assume that authors wanted to say 0.1µM MT. Perhaps, it would be interesting to emphasize in subsection 3.2, that MT restored the percentage of pb1 extrusion, embryo cleavage and blastocysts to similar levels to those observed in the control group.
Response: Thank you for the excellent reminder. We have corrected “1µM MT” to “0.1µM MT” in the revised manuscript.
Figure legend 2 should be shortened. L239 and L245 can be removed.
Response: Thank you for the good comments. The indicated sentences have been removed in the revised manuscript.
In section 3.3, L254-256 and then L256-258 sounds repetitive. Perhaps the authors could unify both sentences and reorganize figure 3. The significant differences in the proportion of misaligned chromosomes and aberrant spindles between control, DT and DT+MT groups can be described first (fig 3B and 3C) and then the authors can say that representative images about the changes in chromosome and spindle organization can be observed in figure 3A (to support the results of fig 3B and 3C).
Response: Thank you for the good comments. The indicated sentences are logically reasonable in the manuscript. Firstly, we introduced the morphology of normal chromosome and spindle in oocytes in Figure 3A, and then described whether DT exposure disrupted the morphology of chromosome and spindle, and MT supplementation restored the morphology of chromosome and spindle in oocytes.
I didn’t find the protocol of polymerization of actin filaments in material and methods, this part should be added. L293-294 corresponds again to mat and methods.
Response: Thank you for the good comments. The protocol of polymerization of actin filaments belongs to immunofluorescence staining so that we did not wrote the part in material and methods.
L314-315 should be removed.
Response: Thank you for the good comment. The sentence is useful for understanding the results. Hopefully that the sentence could be kept in the manuscript.
L357-359 should be removed or added to discussion.
Response: Thank you for the good comment. The introductory sentence of every subsection can help readers understand the aim of the experiments.
L360-361, please add compared to the control and MT group (P < 0.05; Fig. 6A and 6B) and removed L361-364 to avoid repetition. By adding this, you can also shorten this paragraph.
Response: Thank you for the good suggestion. We have modified the sentences in the revised manuscript as suggested.
L365 should be added to material and methods and explain better the subsection 2.3 of immunostaining. Perhaps, this subsection of material and methods can be divided and explain separately the protocol of evaluating chromosome and spindle organization, actin polymerization, DNA damage and autophagy.
Response: Thank you for the good comments. The introductory sentence of every subsection can help readers understand the aim of the experiments. The same protocol for evaluating chromosome and spindle organization, actin polymerization, DNA damage and autophagy should be unified.
Why do you use γH2A.X for detecting DNA damage instead of the TUNEL assay?
Response: Thank you for the good comment. γH2A.X staining indicates the status of DNA breakdown whereas TUNEL staining demotes the status of cellular apoptosis. Hence, γH2A.X staining was used to detect DNA damage.
L370, please add compared to the DT group.
Response: Thank you for the good comment. We have added “compared to the DT group” in the revised manuscript.
L385-386 should be removed or added to discussion.
Response: Thank you for the good comment. The introductory sentence of every subsection can help readers understand the aim of the experiments.
L387-388 should be removed or added to materials and methods.
Response: Thank you for the good comment. The indicated sentences have been removed in the revised manuscript.
L389, please add ‘’ We found that DT exposure led to a significant increase in protein levels of LC3, an autophagy marker, compared to control group.
Response: Thank you for the good suggestion. We have modified the indicated sentences in the revised manuscript as suggested.
DISCUSSION
In general, this section needs improvements. A better connection of all the results obtained in the study is necessary. I suggest connecting all the results that can be responsible for the poor meiotic maturation by the end of the discussion and emphasize how the lower meiotic and cytoplasmatic maturation of DT oocytes as well as their lower developmental competence affects the embryo and blastocyst rate.
Response: Thank you for the good comments. We need to provide deep discussion for each result. All results are logically connected together. The abnormal phenotype of nuclear and cytoplasmic maturation in DT-exposed oocytes should be attributed to the aberrant distribution of cytoskeleton, CGs, and mitochondria. The poor maturation of nucleus and cytoplasm in DT-exposed oocytes could account for the lower developmental competence of embryos. Altogether, the present discussion is sufficient to explain the results observed in this study.
I missed some comparisons of the results obtained in oocyte maturation and its developmental competence with similar papers using another environmental pollutant, specially about the role of melatonin.
Response: Thank you for the good comments. The indicated discussions have been emphasized in the last paragraph of discussion section.
L402-407. This first paragraph is quite similar to the introduction. Therefore, it sounds repetitive and can be avoided.
Response: Thank you for the good comment. This first paragraph mainly aims to propose the scientific question in this study and makes a brief summary of all results. The first paragraph will lay a basis for the subsequent discussions.
L428, what is the role of CGs in oocyte cytoplasmic maturation?
Response: Thank you for the good comment. The translocation of cortical granules during meiotic maturation and proteins stored in cortical granules are necessary for oocyte cytoplasmic maturation. We have added the information in the revised manuscript.
L432-433 should be rewritten, I imagine that you are saying that a normal distribution of CGs is required to block polyspermy, but it is not well written.
Response: Thank you for the good comment. We have rewritten the sentence in the revised manuscript as suggested.
In the last paragraph, it would be interesting to highlight the beneficial effects of MT on embryo cleavage rate and blastocysts, which also proves that MT restores oocyte maturation and their developmental competence.
Response: Thank you for the good comment. We have highlighted the beneficial effects of MT on early embryonic development in the revised manuscript. Also, we added some references to emphasize the protective effects of MT on developmental competence of environmental pollutant-exposed oocytes.
Round 2
Reviewer 4 Report
.